# Characterizing Sampling and Quality Screening Biases in Infrared and Microwave Limb Sounding

Luis F. Millán[1], Nathaniel J. Livesey[1], Michelle L. Santee[1], and Thomas von Clarmann[2]

[1]Jet Propulsion Laboratory, California Institute of Technology, Pasadena, California, USA
[2]Institut für Meteorologie und Klimaforschung, Karlsruhe Institute of Technology, Karlsruhe, Germany

*Correspondence to:* L. Millán (luis.f.millan@jpl.nasa.gov)

**Abstract.** This study investigates orbital sampling biases and evaluates the additional impact caused by data quality screening for the Michelson Interferometer for Passive Atmospheric Sounding (MIPAS) and the Aura Microwave Limb Sounder (MLS). MIPAS acts as a proxy for typical infrared limb emission sounders, while MLS acts as a proxy for microwave limb sounders. These biases were calculated for temperature and several trace gases by interpolating model fields to real sampling patterns and, additionally, screening those locations as directed by their corresponding quality criteria. Both instruments have dense uniform sampling patterns typical of limb emission sounders, producing almost identical sampling biases. However, there is a substantial difference between the number of locations discarded. MIPAS, as a mid-infrared instrument, is very sensitive to clouds, and measurements affected by them are thus rejected from the analysis. For example, in the tropics, the MIPAS yield is strongly affected by clouds, while MLS is mostly unaffected.

The results show that upper tropospheric sampling biases in zonally averaged data, for both instruments, can be up to 10% to 30%, depending on the species, and up to 3 K for temperature. For MIPAS, the sampling reduction due to quality screening worsens the biases, leading to values as large as 30% to 100% for the trace gases and expanding the 3 K bias region for temperature. This type of sampling bias is largely induced by the geophysical origins of the screening (e.g. clouds). Further, analysis of long-term time series reveals that these additional quality screening biases may affect the ability to accurately detect upper tropospheric long-term changes using such data. In contrast, MLS data quality screening removes sufficiently few points that no additional bias is introduced, although its penetration is limited to the upper troposphere while MIPAS may cover well into the mid troposphere in cloud-free scenarios. We emphasize that the results of this study refer only to the representativeness of the respective data, not to their intrinsic quality.

## 1 Introduction

Satellite limb sounders have provided a wealth of information for studies affecting climate, ozone layer stability, and air quality, as well as evaluation of reanalyses and chemistry climate models. Compared to ground-based instruments or aircraft field campaigns, satellite data provides continuous coverage over large areas (or even global scales, depending on their sampling), facilitating model evaluation on a large scale. Further, like many ground-based data sets, satellite missions such as the Atmo-

spheric Chemistry Experiment Fourier Transform Spectrometer (ACE-FTS) (Bernath et al., 2005) and the Aura Microwave Limb Sounder (MLS) (Waters et al., 2006) have records that span more than a decade. In addition, data records constructed using several satellite instruments that span more than 3 decades (Froidevaux et al., 2015; Davis et al., 2016) provide the opportunity to study and evaluate long-term variability and trends. However, satellite observations sample the continuously changing atmosphere only at discrete locations and times, which can result in a biased depiction of the atmospheric state.

Several studies have evaluated the impact of orbital sampling by comparing raw model fields against satellite-sampled ones (e.g., McConnell and North, 1987; Bell and Kundu, 1996; Engelen et al., 2000; Luo et al., 2002; Brindley and Harries, 2003; Aghedo et al., 2011; Guan et al., 2013). For the limb sounding technique, Sofieva et al. (2014) estimated the sampling biases in zonal mean $O_3$ profiles from six limb-viewing satellite instruments and proposed a simple parameterization to estimate them. Toohey et al. (2013) characterized the sampling bias for $H_2O$ and $O_3$ for 16 satellite instruments, including limb scattering sounders, solar and stellar occultation instruments and limb emission sounders. They concluded that coarse non-uniform sampling leads to non-negligible zonal mean biases, not only through non-uniform spatial sampling but mostly through non-uniform temporal sampling, that is, producing means using measurements that span less than the full period in question. Millán et al. (2016) studied the sampling bias for temperature and several trace gas species for a subset of the instruments used by Toohey et al. (2013) and investigated the impact of such biases upon stratospheric trend detection. They found that coarse non-uniform sampling patterns can induce significant errors in the magnitudes of trends inferred directly from monthly zonal means, necessitating analysis of considerably more years of data to conclusively detect a trend. In contrast, dense uniform sampling patterns accurately reproduce the magnitude of the trends, with the number of years of data required determined mostly by natural variability.

However, none of these studies have quantified the additional biases introduced through quality screening of the measurements, that is, this study isolates and quantifies another source of uncertainty in averaged data. Many of the measurements discarded through quality screening have been affected by the presence clouds, which pose a substantial challenge to limb observations as the long limb path traverses hundreds of kilometers. The impact of such cloudy scenes depends on the measurement technique used. For example, instruments measuring microwave emission are unaffected by all but the largest particles in the thickest clouds. Many other limb measurements are screened out because of temperature gradients near the poles, whose impact varies depending on the retrieval scheme, i.e., one dimensional versus tomographic, as well as how accurately the a priori or initial guess captures such gradients (e.g., Livesey and Read, 2000; Carlotti et al., 2001, 2006; Kiefer et al., 2010; Castelli et al., 2016).

This study examines the sampling bias and quantifies the impact of quality screening upon two limb viewing instruments, one using microwave emission (the Aura Microwave Limb Sounder - MLS) and the other one using infrared emission (the ENVISAT Michelson Interferometer for Passive Atmospheric Sounding - MIPAS). Both instruments have dense uniform sampling distributions, which should minimize the sampling biases; however, there is a substantial difference in the number of measurements rejected through quality screening for these techniques (see Figure 1). The discarded profiles tend to cluster geophysically, leading to biases in analyses that are based on the remaining measurements. We emphasize that the results of this study refer only to the representativeness of the respective data, not to their intrinsic quality. Their quality has been exten-

sively evaluated in numerous data characterization and validation papers (i.e., Pumphrey et al., 2007; Read et al., 2007; Santee et al., 2007; Schwartz et al., 2008; Stiller et al., 2012; Hegglin et al., 2013; Raspollini et al., 2013; Neu et al., 2014; Livesey et al., 2017; Sheese et al., 2017). Furthermore, their long-term stability has also been studied (i.e., Nair et al., 2012; Eckert et al., 2014; Hubert et al., 2016; Hurst et al., 2016).

## 2 Data and Methodology

### 2.1 Model Fields

CMAM30-SD (the Canadian Middle Atmosphere Model run in Specified Dynamics mode) is a coupled chemistry climate model nudged to the winds and temperatures of the ERA-Interim reanalysis. This nudging exploits the much better dynamics of the reanalysis to reliably predict the chemical fields. More information can be found in Scinocca et al. (2008), de Grandpré et al. (2000) and McLandress et al. (2014). Extensive validation (de Grandpré et al., 2000; Hegglin and Shepherd, 2007; Melo et al., 2008; Jin et al., 2005, 2009) has shown that the free-running version of this model performs well against observations relevant to dynamics, transport, and chemistry. Comparisons against ACE-FTS and the Odin Optical Spectrograph and Infrared Imaging System (OSIRIS) have shown that CMAM30-SD has a good representation of stratospheric temperature, $H_2O$, $O_3$, and $CH_4$ in polar regions (Pendlebury et al., 2015). Further, CMAM30-SD has been used to construct a long-term $H_2O$ record, acting as a transfer function between satellite observations (Hegglin et al., 2014), and it reproduces halogen-induced midlatitude $O_3$ depletion sufficiently well to be used in long-term $O_3$ trend studies (Shepherd et al., 2014).

The CMAM30-SD version used in this study has a horizontal resolution of approximately $3.75°$, that is, approximately $400\,km$ (similar to the $\sim 500\,km$ limb viewing path length). It has a lid at $0.0007\,hPa$ with 63 vertical levels that vary from $\sim 500\,m$ in the lower troposphere to $\sim 3\,km$ in the mesosphere. Here, we present results using the $H_2O$, $O_3$, CO, $HNO_3$, and temperature CMAM30-SD fields. Note that for this study it is not necessary for the model fields to be correct in absolute terms. CMAM30-SD is simply used as a representative evolving atmospheric state.

### 2.2 Satellite Instruments

We analyze the impact of sampling and quality screening of the limb emission sounders MIPAS and MLS. MIPAS (Fischer et al., 2000, 2008) was launched in March 2002 on the European Space Agency Environment Satellite. MIPAS was a Fourier transform spectrometer conceived to record limb emission spectra. It covered the mid-infrared region from 685 to $2410\,cm^{-1}$ in five spectral bands, allowing retrievals of temperature, pressure and trace gases. MIPAS measured around 1350 vertical scans daily, providing global observations.

From July 2002 to March 2004, MIPAS operated in full resolution mode, with a spectral spacing of $0.025\,cm^{-1}$; however, following persistent malfunctions with the interferometer slide mechanism, instrument operations were temporarily suspended. In January 2005 operations were resumed with MIPAS operating at a spectral spacing of $0.0625\,cm^{-1}$. This mode of operation is known as optimum resolution, and it is characterized by finer vertical and horizontal sampling attained through the degraded

spectral spacing. MIPAS took quasi-continuous measurements until April 2012, when the European Space Agency lost contact with ENVISAT.

In the optimum resolution operation, MIPAS has several measurement modes: the nominal mode, with 27 tangent heights from 6 to 70 km; the middle atmosphere mode, with 29 tangent heights from 18 to 102 km; and the upper atmosphere mode, with 35 tangent heights from 42 to 172 km. The nominal mode covers the entire stratosphere extending into both the upper troposphere and the lower mesosphere to study linkages between these atmospheric layers. In this study we use the geolocations of this measurement mode because it covers around 80% of the measurement time (Fischer et al., 2008).

Several retrieval algorithms have been developed for the MIPAS spectra (e.g. Ridolfi et al., 2000; von Clarmann et al., 2003; Hoffmann et al., 2005; Carlotti et al., 2006; von Clarmann et al., 2009; Dudhia, 2017). Here we use the profiles generated by the Institute of Meteorology and Climate Research (IMK) in cooperation with the Instituto de Astrofísica de Andalucía (von Clarmann et al., 2009) in particular version 5. MIPAS IMK/IAA algorithm retrieves temperature and more than 30 species including $O_3$, $H_2O$, CO, CFCs, PAN, among many others. This retrieval algorithm uses a Tikhonov regularization; it is capable of handling deviations from local thermodynamic equilibrium, and it includes temperature horizontal gradients along the line of sight to prevent many retrievals from failing to converge, particularly near the boundary of the poles. Horizontal gradients are important because the line of sight extends around a thousand kilometers, crossing situations where the assumption of horizontal homogeneity is unrealistic (i.e., 1D retrievals). Several studies have discussed the advantages of including inhomogeneities along the line of sight for atmospheric retrievals (e.g., Livesey and Read, 2000; Carlotti et al., 2001, 2006; Kiefer et al., 2010; Castelli et al., 2016).

MLS (Waters et al., 2006) was launched in July 2004 on the Aura spacecraft. MLS measures limb millimetre and submillimetre atmospheric thermal emission in spectral regions near 118, 191, 240, and 640 GHz, and 2.5 THz. These radiances are inverted using a 2D tomographic optimal estimation algorithm (Livesey et al., 2006) that allows the retrieval of temperature and composition. MLS measures around 3500 vertical scans daily, providing near-global (82°S to 82°N) observations.

To investigate the impact of sampling and quality screening, daily CMAM30-SD model fields were linearly interpolated to the actual latitude and longitude of the satellite measurements. For the MIPAS sampling pattern, for each calendar day of the year we identify the year with the most measurements obtained on that date. That is to say, for the 1st of January, we use the locations of the 1st of January for the year with the most measurement locations, etc. This allows us to have a complete year of MIPAS measurements without interruptions due to MIPAS changing measurement modes. For MLS, we use 2008 as a representative year. To avoid differences attributed to diurnal cycles, all satellite measurements were assumed to be made at 12:00 UT on a given day, avoiding any interpolation in time. Further, we used the vertical grid of the CMAM30-SD fields; that is, the impact of the vertical resolution of the measurements is not taken into account. However, note that, in this case, both instruments have similar vertical resolutions in the upper troposphere / lower stratosphere (UTLS), varying overall from 3 to 4 km.

We constructed three time series: one using the raw CMAM30-SD fields; another using all the measurement locations available; and lastly one using only the measurement locations remaining after the quality screening recommended for each instrument was applied, in other words, after those points flagged as bad values in the actual data were eliminated. The screening

procedure applied to MIPAS data is as follows: We neglect profile points where the diagonal element of the averaging kernel is less than 0.03 — to avoid retrievals influenced by the a priori — and discard points where the visibility flag was set to zero — which indicates that MIPAS has not seen the atmosphere at those particular altitudes. The screening procedure applied to MLS data follows the guidelines detailed by Livesey et al. (2017): We only use data within the specified pressure ranges; we neglect profile points for which the precision is negative (which indicates that the retrievals are influenced by the a priori); we avoid profiles for which the "Status" field is an odd number (which indicates operational abnormalities or problems with the retrievals); and we only use profiles for which the "Quality" and "Convergence" fields are within the specified thresholds. The "Quality" field describes the degree to which the measured radiances have been fitted by the retrieval algorithm and the "Convergence" field is a ratio of the fit achieved at the end of the retrieval process to the value predicted at the previous step.

Figure 1 shows typical daily MIPAS and MLS geolocations overlaid on top of a modeled water vapor map. Both instruments have dense coverage that, as noted by Toohey et al. (2013), is relatively uniform with latitude and time. Figure 1 also displays those geolocations for which the retrieved values are not recommended for scientific studies, that is, they are screened out by the quality criteria. As shown, these failed or in many cases skipped retrievals cluster in the tropics or near the poles. Overall, in the tropics, these missing retrievals are due to clouds. Near the poles, the retrieval failures are presumably due to temperature horizontal gradients and, in the case of MIPAS, also due to the presence of polar stratospheric clouds. The substantial difference between the number of failed/missing retrievals in the tropics for MIPAS and MLS is the main motivation for this study.

To quantify this further, Figure 2 displays the yield given by,

$$Y = \frac{N_{QS}}{N_A} \tag{1}$$

where $N_A$ is the number of measurements available and $N_{QS}$ is the number of measurements left after applying the quality screening criteria at each latitude and each pressure level. Again, MIPAS low yield values accumulate in the tropics and near the poles. Overall, MIPAS yields drop below 60% near the South Pole at pressures greater than $\sim$20 hPa and below 30% in the tropics at pressures greater than 100 hPa. In contrast, in general MLS yield values are better than 90%, although the measurements do not extend below the upper troposphere. The two exceptions are the yield values for $H_2O$ near the South Pole, which drop below 90%, and $HNO_3$ near the equator, which drop to 60%. Note that MIPAS yield values drop below 10% well into the troposphere.

## 3 Induced Sampling and Quality Screening Biases

Following Millán et al. (2016), we evaluate the sampling biases as well as the quality screening biases associated with constructing monthly zonal means using the raw CMAM30-SD fields, $Z_R$, versus those using the satellite-sampled measurements, $Z_A$, or only those passing the quality screening criteria, $Z_{QS}$. The difference between $Z_A$ or $Z_{QS}$ and $Z_R$ gives the sampling or the quality screening induced bias, respectively. For each instrument and for each month throughout one year, we computed these biases as a function of latitude and pressure. Note that the quality screening bias is the sampling bias plus the additional impact of screening out more locations and, hence, reducing the sampling frequency.

Figure 3 shows examples of the sampling and screening biases for June 2005. Percentage biases are shown for the trace gases to cope with their large vertical variability. MIPAS and MLS sampling biases are practically identical. For the trace gases, sampling biases are larger in the upper troposphere, where the variability is larger, while the temperature sampling biases are larger near the edges of the polar regions, where there are substantial temperature gradients. The impact of the MIPAS quality screening is evident in the tropics (in particular near 20°N), where the yield values are expected to be greatly affected (see Figure 2) by clouds. In this region, on top of the sampling biases, all parameters studied display an underestimation, for example, up to -50% for $H_2O$. Although this resembles the expected dry bias in clear-sky tropospheric infrared measurements (e.g., Sohn et al., 2006; Yue et al., 2013) — that is, the fact that infrared instruments cannot measure cloudy regions where $H_2O$ is high, resulting in a dry bias — the biases shown here are due to a combination of two factors: (1) high $H_2O$ values associated with deep convection (the screened-out locations might not necessarily be cloudy in the model fields however they are for the most part in the tropics, in regions of high $H_2O$ values, see Figure 1 for an example) and (2) due to the reduced sampling frequencies. Note that this is also applicable to other parameters; that is, the quality screening biases shown here are not an indication of trace gas (or temperature) / deep convection relationships. In contrast to those of MIPAS, except for the reduced vertical ranges, MLS sampling biases are unaffected by data quality screening.

We note that the cloud screening procedure of the IMK/IAA algorithm is conservative with respect to those of other MIPAS processors (Spang et al., 2012). On the face of it, it appears that this causes an unnecessarily large sampling bias which could be avoided by using a less restrictive cloud screening threshold value. The purpose of a conservative cloud screening procedure is to guarantee that the measurements passing the cloud screening are indeed unaffected by any cloud signal in the spectra. Cloud signals can lead to systematic retrieval errors, which are correlated to the state of the atmosphere, hence, the sampling biases would merely be replaced by retrieval biases. This, we think, is worse, because then both the parent data and the zonal averages would be affected, while with a conservative screening only the averages are affected but not the parent data which survive the screening.

To summarize the potential sampling and quality screening biases, Figure 4 shows their root-mean-square (RMS) computed over one year's worth of data. Again, the MIPAS and MLS RMS sampling biases are almost identical: $H_2O$ displays a bias of up to 30% at pressures greater than ~150 hPa; CO, $O_3$ and $HNO_3$ show biases (up to 30% for $O_3$ and $HNO_3$) near mid-latitudes (around 40°S and 40°N), where there are sharp trace gas gradients and variability due to tropopause folding; and temperature displays a bias as large as 3 K near the polar edges.

The impact of MIPAS quality screening is especially evident in $H_2O$ and $HNO_3$, which have potential biases as large as 100%, but quality screening also affects the rest of the parameters: CO and $O_3$ biases approach 30% in the tropics, while the region with 3 K temperature bias expands near the South Pole. As before, except for the reduced vertical ranges, the impact of the MLS quality screening is negligible; that is, the screening biases are almost identical to the sampling biases.

To exemplify the impact of these quality screening biases, Figure 5 (left) shows time series (1979-2012) of 20°S to 20°N $H_2O$ at 200 hPa using the raw CMAM30-SD fields, the full satellite-sampled fields and only those points passing the screening criteria. All time series show the expected features, with an annual cycle related to the seasonality of the cold point tropopause temperature. The MIPAS time series constructed using the full satellite-sampled fields is almost identical to the one constructed

using the raw CMAM30-SD field. However, as suggested by the screening bias shown in Figure 4, the MIPAS time series using the quality screening displays a substantial dry bias. In contrast, no evidence of such a bias is seen in the MLS time series; that is, both the time series constructed using the full satellite-sampled field and that based on only those points passing the screening criteria are almost identical to the CMAM30-SD one.

Figure 5 (right) shows the area-weighted scatter between these time series. MIPAS sampling scatter, that is, the scatter between MIPAS when using all available measurements and the raw CMAM30-SD fields, is small and their correlation tight, with a bias better than -1.5%, a slope of ∼1.05, and a coefficient of determination of 0.98. The contrast with the MIPAS screened scatter is dramatic in this particular latitude/pressure region; it displays considerably more scatter and, as in the time series (Figure 5-left), a discernible bias. Quantitatively, MIPAS screened data displays a bias of 16.13%, a slope of 1.32, and a coefficient of determination of ∼0.8 (which implies that 20% of the total variation cannot be explained). MLS sampling and screened scatterplots are almost the same.

To explore this further, Figure 6 shows these metrics versus pressure using different latitude bands for the MIPAS sampling scatter. As shown, the coefficients of determination as well as the slopes are close to one and the biases close to zero in most cases. The most notable exceptions are the biases between 20° and 45° (either north or south) for $O_3$ and $HNO_3$, which can be up to -10%. In these regions, Figure 3 and Figure 4 indicate biases due to the sharp trace gas gradients associated with tropopause folding. Note that both the MLS sampling and the MLS screened scatter are almost identical to the MIPAS sampling scatter and, hence, are not shown.

The MIPAS screened scatter results are shown in Figure 7. The largest impact can be found in the tropics (the 20°S-20°N latitude band) at pressures greater than 100 hPa. Here, the coefficients of determination, the biases and the slopes are severely degraded. The coefficients of determination rapidly decrease, especially for $H_2O$, $O_3$ and $HNO_3$, whose values are as low as 0.5 at 200 hPa and worsen further at lower altitudes. The biases for $O_3$ and $HNO_3$ oscillate between -10% and 10% and can be as large as 40% for $H_2O$. Lastly, all the slopes vary from 0.5 to 1.5, depending on pressure level. These poor metrics imply that any trends derived at these pressure levels will also be impacted by quality screening induced biases: the magnitude of the trends will be affected because of the change in the slope, and the number of years of observations required to conclusively detect trends will considerably increase due to the noise associated with the worsening of the coefficients of determination (e.g., Millán et al., 2016). As an example, Figure 8 shows the $H_2O$ and $O_3$ trends in the tropics computed using monthly zonal mean deseasonalized anomalies of the raw model fields, as well as using all the available satellite-sampled measurement locations and only those passing the screening criteria in the tropics. As shown, when all available measurement locations are used, the MIPAS and MLS sampling allows accurate derivation of trends, with values matching those calculated from the raw model fields almost exactly. However, when only those measurements passing the screening criteria are used, both instruments have limitations: MIPAS trends are impacted because of the large percentage of measurements screened out below 100 hPa, which introduces non-negligible artifacts (up to 80% change for $H_2O$ and up to 20% change for $O_3$); MLS trends are impacted because of the reduced vertical resolution, which limits its usefulness to the upper troposphere and above. Note that the impact of quality screening on MIPAS trends can be mitigated by using a regression model similar to the ones used by Bodeker et al. (2013) and Damadeo et al. (2014). These models have been shown to mitigate the effects of the non-uniform temporal,

spatial and diurnal sampling of solar occultation satellite measurements. Furthermore, MIPAS trend analysis can be restricted to regions less affected by deep convection (for example, the mid tropical Pacific) to minimize the quality screening effects.

The estimated number of years required to definitively detect these trends is also shown in Figure 8. These estimates were computed assuming a trend model similar to the one described by Tiao et al. (1990), Weatherhead et al. (1998), and Millán et al. (2016), with a seasonal mean component represented by the monthly climatological means. As shown, with the MIPAS screened fields additional years are required for robust trend detection (up to ∼150 years for $H_2O$ and up to ∼40 years for $O_3$ versus 50 years and 20 years, respectively, when all available measurements are used).

Similar analyses were performed for other latitude bands. Although the magnitude of the trends derived when using the MIPAS screened measurement locations was also impacted in these cases, no significant difference was found in the number of years required to detect such trends. In addition, no significant artifacts were found for HNO3, CO or temperature for either the trend magnitude or the number of years required to detect such trends. Note that, when using real data, the effect of instrument noise upon trends will be negligible due to the vast number of MIPAS or MLS measurements associated with each monthly latitude bin. Drifts and long-term stability issues on these datasets (i.e., Eckert et al., 2014; Hubert et al., 2016; Hurst et al., 2016) will have to be corrected.

## 4   Summary and Conclusions

This study explored the implications of sampling in the UTLS for two satellite instruments, MIPAS and MLS, for $H_2O$, $O_3$, CO, $HNO_3$, and temperature. We quantify sampling biases by interpolating CMAM30-SD fields, used as a proxy for the atmospheric state, to the measurement locations and computing monthly means. Both of these instruments have dense uniform sampling, with around 1350 points spread globally for MIPAS and around 3500 spread from 82°S to 82°N for MLS, resulting in almost identical sampling biases for the two instruments. For the trace gases, the largest sampling biases are found in the upper troposphere, where there is more natural variability: $H_2O$ displays a bias of up to 30%, while CO, $O_3$ and $HNO_3$ show biases near mid-latitudes of up to 10% for CO or 30% for $O_3$ and $HNO_3$ due to sharp trace gas gradients and variability arising from tropopause folding. The temperature sampling bias is negligible (less than 1 K), except near the polar edges, where the bias can be as large as 3 K, presumably due to horizontal temperature gradients.

Besides the orbital sampling biases, this study also evaluated the impact of quality screening, which further reduces the sampling frequency. In the tropics (see Figure 2), MIPAS is substantially impacted by clouds, as they act as grey bodies with high opacity, greatly altering the radiances below the cloud top. Cloud effects are evident, with $H_2O$ and $HNO_3$ biases up to 100% and CO and $O_3$ biases up to 30%. In contrast, because of their longer wavelengths, MLS measurements are unaffected by all but the thickest clouds, negligibly impacting the sampling frequency. However, continuum absorption in the microwave suppresses signals from the mid and lower troposphere in a limb viewing geometry, limiting the MLS vertical range to the upper troposphere and above while MIPAS may cover well into the mid troposphere in cloud free scenes.

Analysis of scatterplots of time series constructed using the raw model fields versus those using all the available measurement locations (either for MIPAS or MLS) reveal that at most pressure levels and most latitude bands, the coefficient of determination

and the slope of the fits are close to one, while the biases are close to zero. However, when only those measurements passing the screening criteria are used, MIPAS upper tropospheric measurements are severely impacted in some regions. In the tropics, the coefficients of determination rapidly decrease, especially for $H_2O$, $O_3$ and $HNO_3$, from $\sim$1 at 100 hPa to as low as 0.5 at 200 hPa, and they worsen further at lower altitudes. The biases for $O_3$ and $HNO_3$ oscillate between -10% and 10% and can be as large as 40% for $H_2O$. Lastly, all the slopes vary from 0.5 to 1.5, depending on pressure level. These biases affect trends derived from these measurements using a simple regression upon monthly zonal mean data substantially affected by clouds. Further, the number of years required to detect such trends may increase due to the extra noise added to the time series by screening out measurements. Note that although these results were derived for MIPAS, they are applicable to other instruments with dense sampling but for which quality screening (e.g., for clouds) severely impacts their yield.

## 5 Data availability

The datasets used in this study are publicly available: CMAM30-SD fields can be found in the Canadian Centre for Climate Modeling and Analysis webpage (http://www.cccma.ec.gc.ca/data/cmam/output/CMAM/CMAM30-SD/index.shtml), MLS data can be found in the NASA Goddard Space Flight Center Earth Sciences Data and Information Services Center (http://disc.sci.gsfc.nasa.gov/holdings/MLS/index.shtml), and MIPAS data can be found in the Karlsruhe Institute of Technology webpage (https://www.imk-asf.kit.edu/english/308.php).

*Acknowledgements.* Work at the Jet Propulsion Laboratory, California Institute of Technology, was done under contract with the National Aeronautics and Space Administration. We thank David Plummer of Environment Canada for his assistance in obtaining the CMAM30-SD dataset. We thank the reviewers for their useful comments. Government sponsorship acknowledged.

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

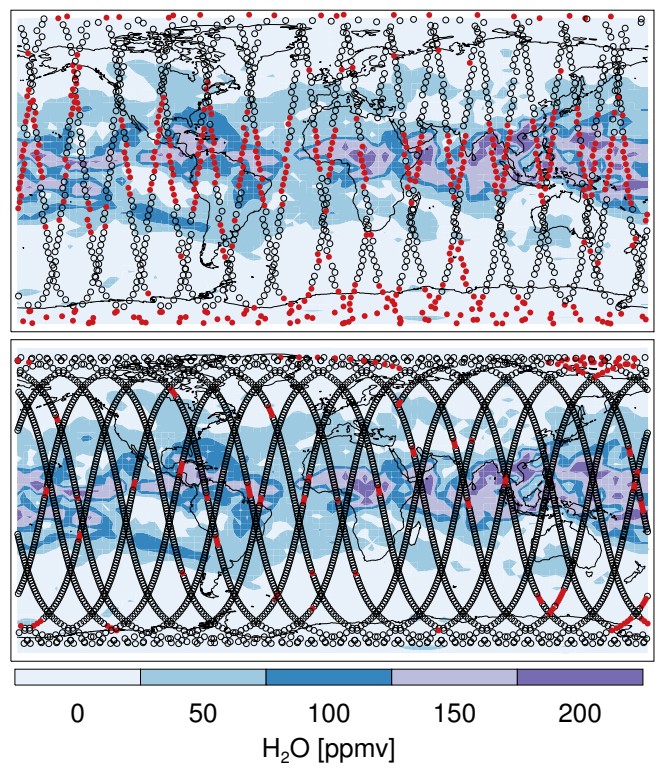

**Figure 1.** Typical MIPAS (top) and MLS (bottom) sampling overlaid on top of a modeled water vapor map (June 1st, 2005) at 200 hPa. Red dots show missed or failed retrievals: in the tropics, these are mostly due to clouds.

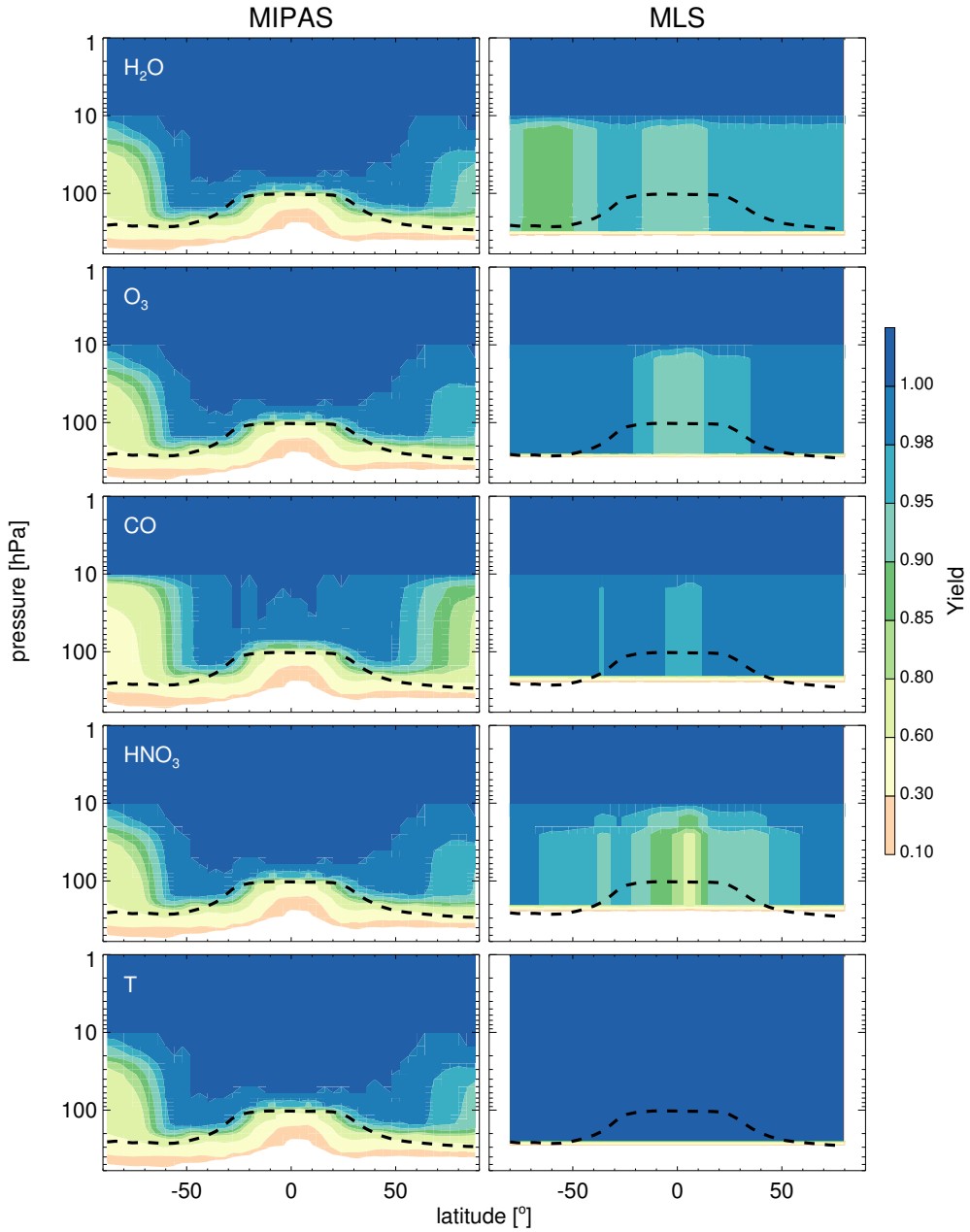

**Figure 2.** MIPAS and MLS zonal mean yield ($N_{QS}/N_A$ where $N_A$ is the number of measurements available and $N_{QS}$ is the number of measurements left after applying the quality screening criteria) for $H_2O$, $O_3$, CO, $HNO_3$ and temperature for 2005, that is, sampling the modeled 2005 year with the sampling patterns as explained in the text. Note the non-linear color scale. The dashed black lines show the mean 2005 thermal tropopause derived from the Modern Era Retrospective Analysis for Research and Application-2 (MERRA2) fields (Bosilovich et al., 2015). This tropopause information was obtained from the derived meteorological products as described by Manney et al. (2007, 2011).

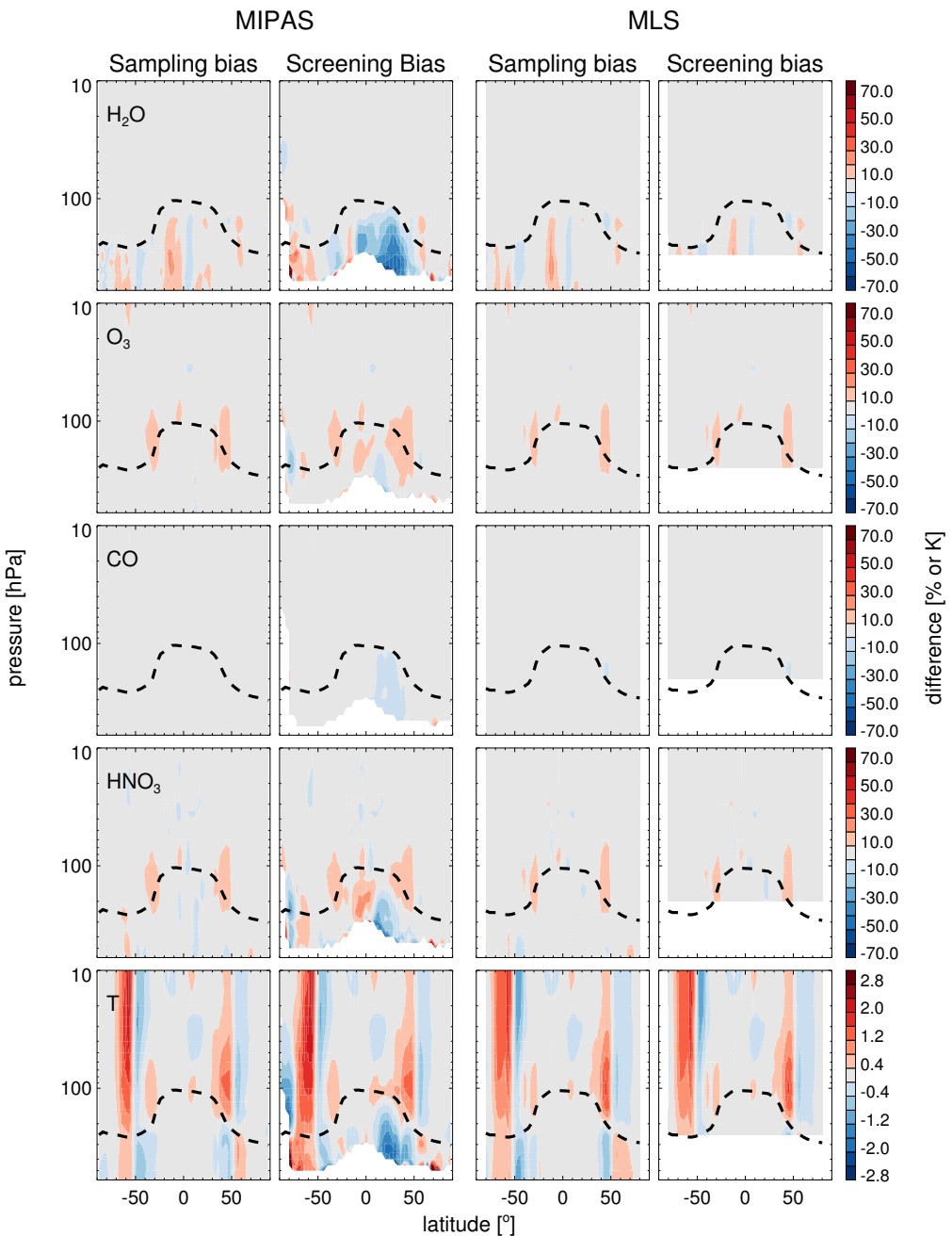

**Figure 3.** June 2005 sampling and quality screening biases as a function of latitude and pressure for $H_2O$, $O_3$, CO, $HNO_3$, and temperature as measured using MIPAS and MLS. White regions denote a lack of measurements. The dashed black lines show the mean 2005 thermal tropopause derived from MERRA2.

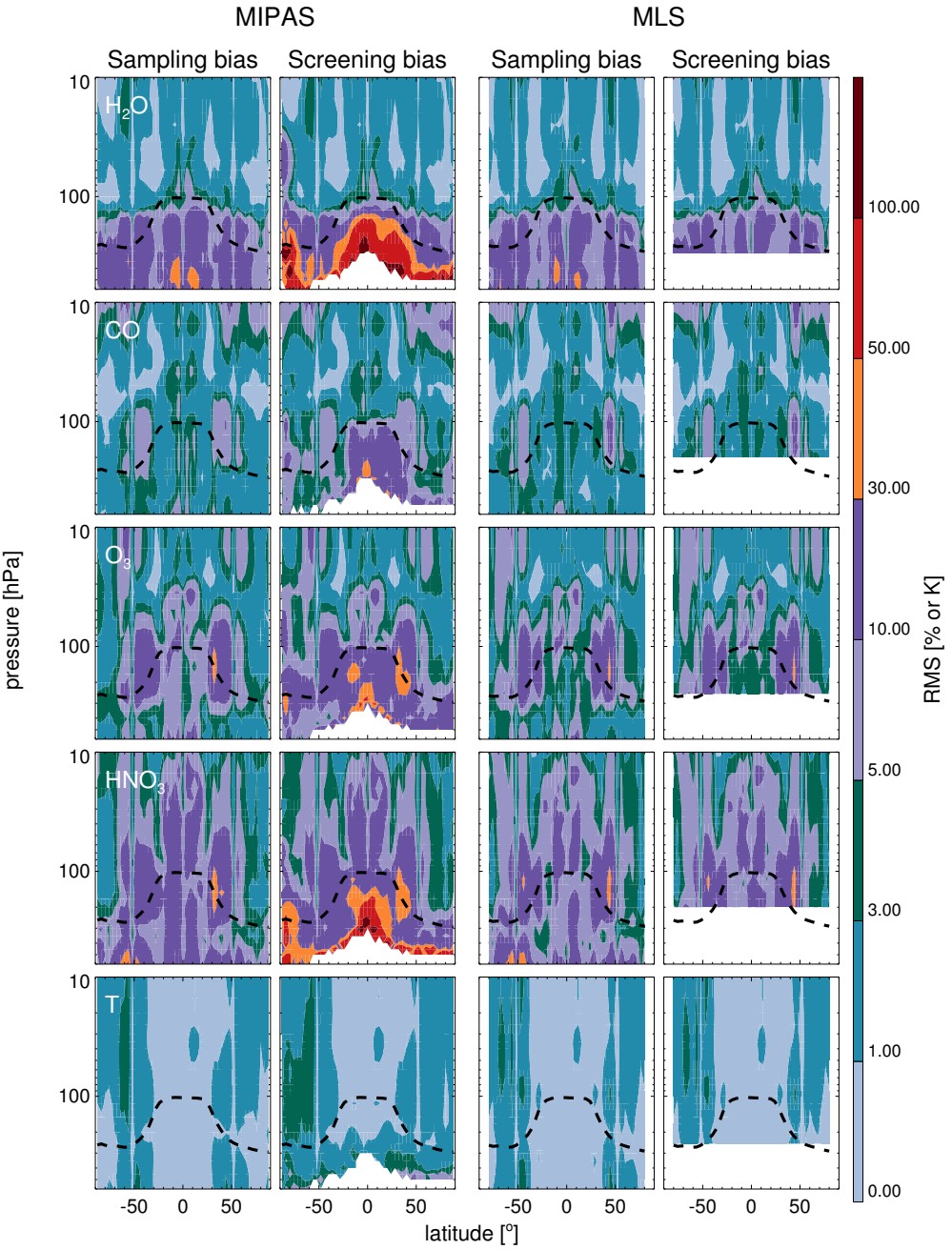

**Figure 4.** Root-mean-square sampling and quality screening biases for 2005 as a function of latitude and pressure for $H_2O$, $O_3$, $CO$, $HNO_3$, and temperature as measured using typical MIPAS and MLS data coverage. The dashed black lines show the mean 2005 thermal tropopause derived from MERRA2.

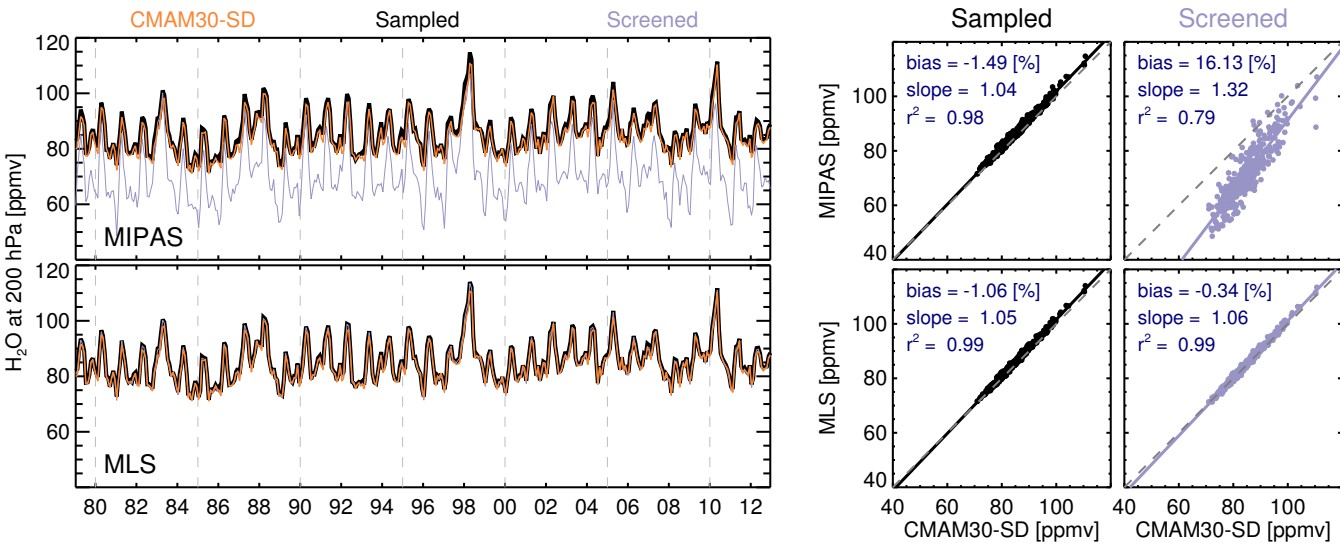

**Figure 5.** (left) Time series of $20°S$-$20°N$ $H_2O$ at 200 hPa for the raw CMAM30-SD fields (orange lines), the full satellited-sampled fields (black lines) and only those points passing the quality screening criteria (thin purple lines) for MIPAS and MLS. (right) Scatterplots between these time series. The dashed gray lines are the 1:1 line, and the solid lines are the linear best fits, whose slopes are given. Also, the coefficient of determination, $r^2$, and the bias are shown.

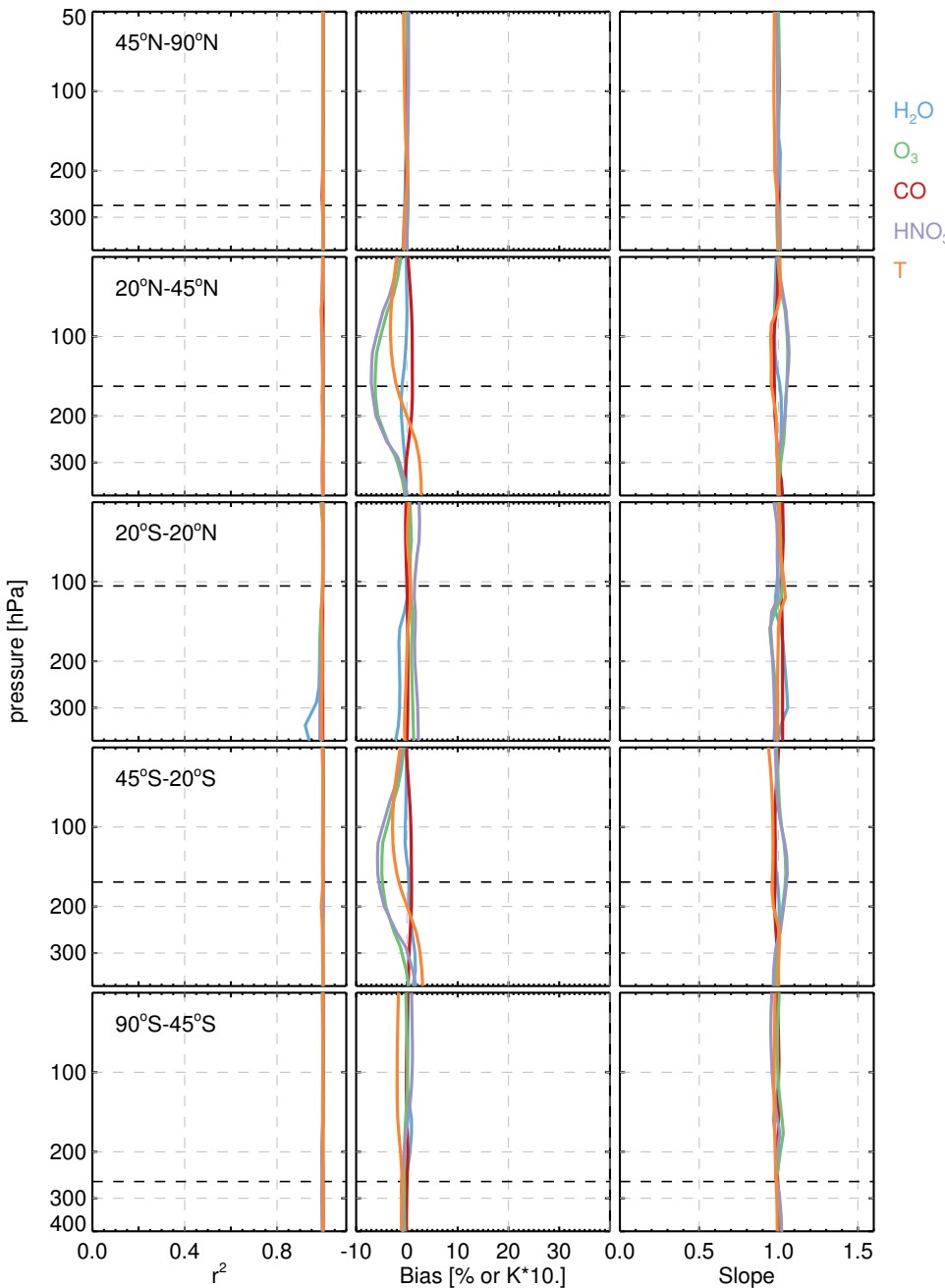

**Figure 6.** Vertical profiles of the coefficient of determination, the bias, and the linear fit slope for different latitude bands for the MIPAS vs CMAM30-SD scatter using the full satellite-sampled fields. Note that for clarity the temperature bias is shown as K*10. Blue, green, red, purple, and orange lines represent $H_2O$, $O_3$, CO, $HNO_3$, and temperature metrics. The dashed black lines show the mean 2005 thermal tropopause derived from MERRA2 for the particular latitude bands.

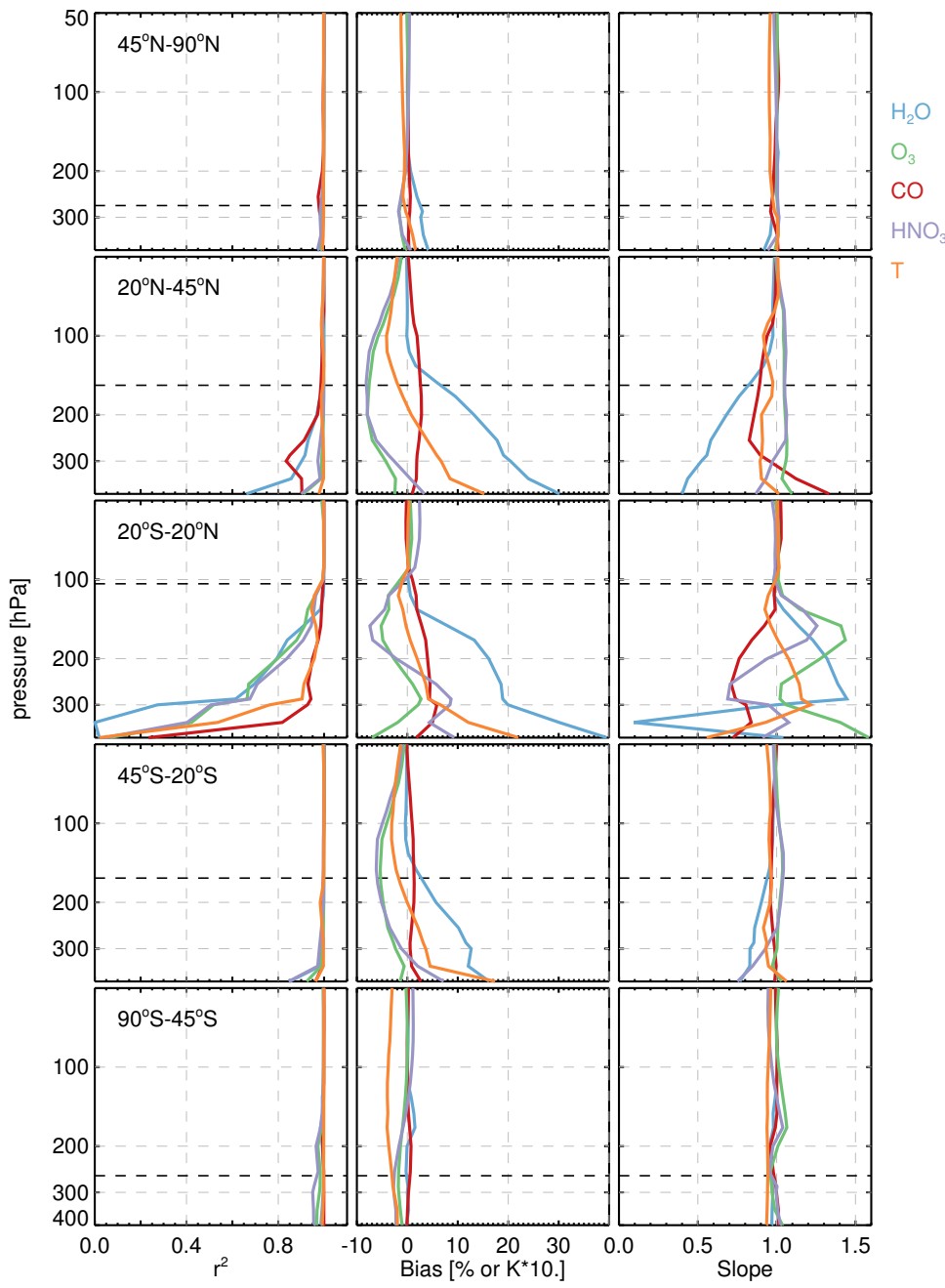

**Figure 7.** As in Figure 6 but using only the profiles that passed the quality screening.

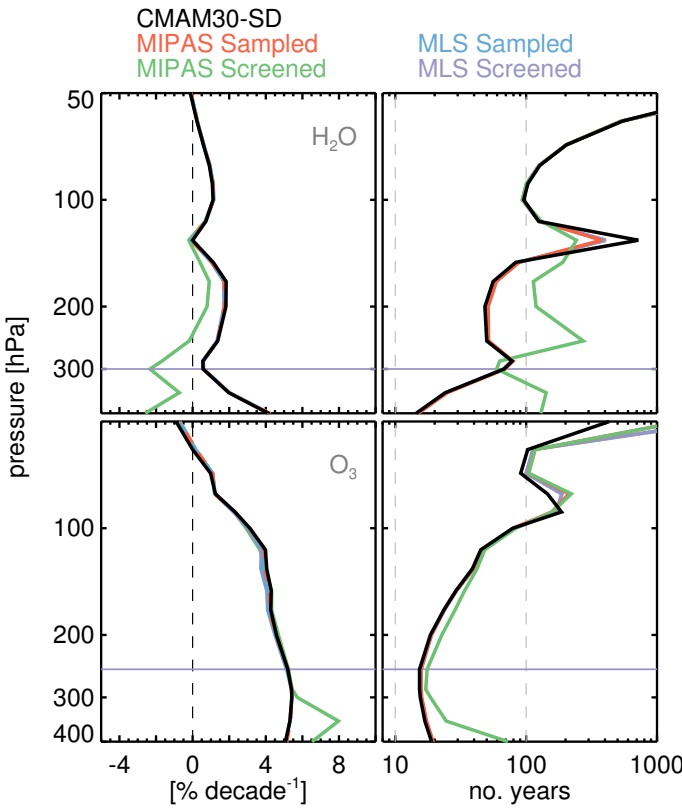

**Figure 8.** (left) $H_2O$ and $O_3$ trends computed based on monthly zonal mean deseasonalized anomalies for the tropics (20°S to 20°N) using the raw model fields, all the available satellite measurements (MIPAS or MLS sampled) and only those measurements passing the screening criteria (MIPAS or MLS screened). Note that for $O_3$, we only use data starting from 2000 to capture the expected period of $O_3$ recovery. A purple line indicates the bottom (largest pressure) of the recommended range of the MLS retrievals. (right) Number of years required to detect such trends.