# Peer review of "Characterizing Sampling and Quality Screening Biases in Infrared and Microwave Limb Sounding"

_Atmospheric Chemistry and Physics, 2017_

## Referee Comment (RC1) · Anonymous Referee #1 · 1 Nov 2017

This paper discusses the impact of non-uniform sampling of the MIPAS and MLS instruments on resulting zonally averaged data with particular emphasis on how data quality screening exacerbates biases. The authors describe how MIPAS and MLS have similar sampling biases but how MIPAS data is detrimentally affected by screening through the use of running MIPAS and MLS sampling through the CMAM30-SD CCM. While the methodology applied has some use and accurate in its basic conclusions, some of the more meaningful conclusions (e.g., impacts on ability to derive trends from different instruments in the UTLS) are not supported by analysis and thus speculative. Furthermore, the scope of this work is severely limited and the tone of the paper is seemingly self-serving. As such, I would recommend this work for publication only after additional work and major revision.

[Figure]

**Major Comments:**

Pg. 02, Line 29: "We emphasize that the results of this study refer only to the representativeness of the respective data, not to their intrinsic quality."

This statement greatly detracts from the value of this work. This work makes very clear remarks regarding the impact of sampling biases on the ability to use certain kinds of data sets for trend analysis in the UTLS and how sampling biases will require longer data sets because of the added noise. However, the quality of the data sets that are used is a critical component to those kinds of analyses. Recent work in the 2014 Ozone Assessment or the SI2N effort (https://www.atmos-chem-phys.net/ special_issue284.html) have shown that sampling biases, while present in trend analyses, are not necessarily the greatest driver of trend uncertainty as data quality issues and instrument drifts are also present. Incorporating data quality into the calculation is necessary, as it is still possible for higher precision data with sampling biases to be more robust for trending work than lower precision data, though it ultimately depends on what those precisions and sampling impacts are.

Pg. 06, Line 32 and Pg. 07, Line 25: "These poor metrics imply that any trends derived at these pressure levels will also be impacted by quality screening induced biases: the magnitude of the trends will be affected because of the change in the slope, and the number of years of observations required to conclusively detect trends will considerably increase due to the noise associated with the worsening of the coefficients of determination (e.g., Millán et al., 2016)."

Biases and trends can be completely independent and the slopes that are referred to here are not trend slopes but correlation slopes. For example, a seasonally dependent dry bias in H2O as shown in Fig. 5 does not guarantee an induced bias in the trend values and so the impact of these sampling biases on trends is not directly addressed by this work. Furthermore, while it is true that increased noise in the data can result in requiring longer duration data records to determine significant trends, neither this study

nor Millan et al. (2016) considers the attribution between data quality and non-uniform sampling. Without additional work to address the impacts of each of these factors, this entire statement is only speculative.

In general, the scope of this work is severely limited. The statement that an instrument's inability to see through clouds will result in less data is obvious while comparing two data sets to ascertain the impact on trends without considering data quality is neglectful. As it stands, this study has limited to no scientific value. However, the underlying concept of this work could be expanded albeit with simulated sampling and uncertainties. For example, the authors could simulate different sampling patterns from different orbit types and sampling frequencies for an expected limb sounder. Running these through model data would allow for comparisons of the impact of different potential sampling patterns. To further test potential data quality screening, utilizing model cloud fields or retrieval limitations of different observation techniques would create another variable to test rather than using the data quality screening of a specific instrument. Lastly, the authors could simulate differing data precisions or vertical resolutions from all of these sensitivity tests and incorporate the resulting uncertainties into trending analyses to determine the impact of all of these variables on the derived trends. This would provide a large solution space to test the impacts of potential measurement systems on their ability to be used for trending analyses in the UTLS region. While these suggestions are extensive and perhaps more comprehensive, some aspect(s) would be necessary to incorporate actual data quality into the authors' data quality screening to back up the claims regarding the impact on trend detection.

Perhaps what is most troubling is the tone of the paper. It seems that the purpose of this paper is to act as a published reference for why a future MLS instrument must be used to ascertain long-term trends of trace gas species in the UTLS. The authors appear to go to great lengths to phrase the message as to assert the "superiority" of MLS measurements over MIPAS at every turn (i.e., pointing out all potential deficiencies in MIPAS data without mentioning any for MLS), and even go as far as attempting

to undermine the usefulness of other current or future measurement systems as evidenced by the last two sentences in the paper. While this type of rhetoric is expected in a proposal, it is not suited for a scientific publication.

**Minor Comments:**

Pg. 01, Line 23: "Further, satellite missions such as . . . have records that span more than a decade." This statement is phrased in such a way as to suggest that no ground station data have records that long. I would suggest revising.

Pg. 02, Line 09: "They concluded that coarse non-uniform sampling leads to non-negligible biases . . ." Biases in what? Is that data biased from the sampling or are the analysis methods not conducive to using data with non-uniform sampling?

Pg. 02, Line 13: "They found that coarse non-uniform sampling patterns can induce significant errors in the magnitudes of inferred trends . . ." Again, is this a flaw in the data or the analysis method?

Pg. 04, Line 12: "MLS measures around 3500 vertical scans daily, providing near-global (82S to 82N) observations." Please also include the geospatial sampling characteristics of the MIPAS instrument in its description.

Pg. 04, Line 20: "Further, we used the vertical grid of the CMAM30-SD fields; that is, we assume that MIPAS and MLS vertical resolution is good enough to resolve these model fields, at least in the upper troposphere / lower stratosphere (UTLS)." What are the vertical resolutions of the model and instruments in the UTLS?

Pg. 06, Line 31 and Pg. 07, Line 24: "The biases for O3 and HNO3 oscillate between -10

Figure 5: It would be better to change the X-axis label interval on the time-series plot to every 5 years.

Figures 6 and 7: There is a lot of unnecessary white space in some of these plots,

though given the desire to maintain consistent axes ranges between the two I can see why Figure 6 has so much white space. That having been said, I still think some reductions can be made to make the results easier to see. Additionally, for whatever reason the line thicknesses appear the same between Figs. 6 and 7 when zoomed out but are much thinner in Fig. 6 when zoomed in. Lastly, what is the bottommost pressure level on the Y-axis?

---

## Referee Comment (RC2) · Anonymous Referee #2 · 5 Nov 2017

The paper is dedicated to the characterization of sampling biases in infrared and microwave limb sounding instruments, with MIPAS and MLS taken as examples. The paper is a continuation of a series of publications on characterization of sampling biases. The new aspect is analyzing the influence of quality screening on data representativeness.

MAJOR COMMENTS

1) It is stated in the abstract that " analysis of long-term time series reveals that these additional quality screening biases may affect the ability to accurately detect upper tropospheric long-term changes using such data" (similar statements are on page 6 and in conclusions) However, the performed analyses are insufficient for such statement. It is rather expected that the screening of cloudy conditions results in biased estimates,

and that the variability might not be represented properly. However, biased estimates, not perfect correlation coefficient with the full time series and R2 do not necessary imply that the long-term trends are inaccurate. Furthermore, if the sampling patterns do not change over time, a large part of sampling uncertainty can be removed in the trend analysis by consideration of deseasonalized anomalies. In order to make such statement on ability of accurate trend detection, the authors should perform trend analysis using the full and sub-sampled datasets and support their statements by quantitative estimates. Another, a simpler solution, is to remove these abovementioned statements on ability to accurately detect trends from the manuscript.

If the authors will decide to extend the analyses, it would be also interesting to investigate the influence of sampling patterns on ability to reproduce the natural cycles.

2) The value of the paper will be increased significantly, if the presented analyses of sampling biases using the modelled data are enhanced with comparison of real experimental data from MIPAS and MLS. Such analyses would illustrate whether the observed biases are explained by sampling patterns.

MINOR COMMENTS

1) P.4, L.5 : Please write the version of the IMK/IAA processor

2) P.2 L.11: The sampling uncertainty has been also discussed in (Sofieva et al., 2014). In this paper, the authors analysed the sampling biases for 6 satellite instruments and proposed a parameterization of sampling uncertainty in monthly zonal mean data.

Sofieva, V. F., Kalakoski, N., Päivärinta, S.-M., Tamminen, J., Laine, M. and Froidevaux, L.: On sampling uncertainty of satellite ozone profile measurements, Atmos. Meas. Tech., 7(6), 1891–1900, doi:10.5194/amt-7-1891-2014, 2014.

3) Figure 5: Please use more distinct colors in scatter plots.

---

## Referee Comment (RC3) · Anonymous Referee #3 · 7 Nov 2017

***Review of the ACPD manuscript:*** *'Characterizing Sampling and Quality Screening Biases in Infrared and Microwave Limb Sounding' by Millan et al.*

The paper by Millan et al. describes effects of sampling and quality screening for two limb sounding instruments (MLS and MIPAS). Based on CCM data the analysis shows significant biases in the zonally averaging of various trace gases and temperature distribution, especially for a MIPAS-like instruments with its cloud-induced quality screening. Biases in the zonal means as well as for trend analyses in the upper troposphere can be quite large, which is an important fact and needs consideration in any trend analysis.

**General comments:**

There are two major concerns regarding the manuscript from Millan et al.:

a) The study highlights a very specific caveat of sampling and screening issues of space borne instruments. The whole approach is very technical, investigating a very obvious problem of the IR limb measurements due to cloud screening, which can have a severe effect on the results especially if you like to analyse trends in the very cloudy upper tropical troposphere. Although this effect is of interest for ACP community, these kind of analysis is not really suited for ACP. The whole manuscript looks more like a technical study or a technical note (for example as prepared for NASA or ESA studies for future satellite sensors) and is not sufficient for a full publication in a journal like ACP. A more in-depth study of the MW/IR limb caveats on trend analyses cloud be publishable in a more technical and instrument-orientated journal like Atmospheric Measurement Techniques.

b) To my mind, this paper does not contribute a substantial add-on to a former study of the authors. The paper by Millan et al. (2016) in ACP describes most of the general effects of sampling pattern, which can be quite similar effects to the quality screening, in the context of a much more detailed study on atmospheric trend analyses. My strong impression is that the actual manuscript misses new concepts and methods and just repeats parts of the Millan et al. (2016) analyses with another combination of satellite instruments.

**Major comments:**

1) P4, line 29: Although the screening is a fundamental issue of the study, there are no details presented for MLS (but some for MIPAS). Please, at least summarize the main topics of Livesey et al. (2006). In addition, more details on the cloud screening of MIPAS would be helpful as well (see next comment).
2) P5, line 1: Please note somewhere here that the MIPAS cloud-screening applied in the IMK product is a very conservative approach with respect to cloud effects in the measured spectra. This might have effects on the presented analysis. Especially, in the tropical upper troposphere high water vapour abundance will result in false-positive cloud detections (e.g. Spang et al. 2012) in the IMK-product and consequently an artificial overestimation in the biases, where other MIPAS level 2 processors might do a better job. This objective should be addressed in the manuscript.
3) The disadvantages of the cloud screening in the tropical UT for the IR measurements are highlighted frequently in the manuscript and are presented by different types of analyses, but with only very limited further information content. In contrast, the caveat of MW sounders to retrieve trace gases below ~300 hPa at any latitude is only briefly mentioned.
   Studies for future ESA missions, like the PREMIER report for mission selection (http://projects.knmi.nl/capacity/PREMIER/SP1324-3_PREMIERr.pdf), show nicely the excellent synergy by a combination of IR and MW (s. Fig. 4.2 of the report), where IR-limb still allows measurements in the mid-troposphere, where MW-limb fails to measure.
4) To my mind the trend analysis section needs to be improved. Although MIPAS tackles to deliver good trend analyses in tropical UT, this is a shortened result. Have you tested if the biases are better, if the trend analyses are restricted to longitude regions where convection is less pronounced (e.g. mid-Pacific, Atlantic regions, excluding the warm pool region, central Africa and South America, where deep convection and cloud-coverage is most pronounced). Local trends in the UT/MT in tropical and subtropical areas would be valuable in principle.
5) I am also wondering why the authors argue that most of the biases are not caused by a lack of observations in convective (cloudy) areas in the tropics but are caused by a reduction in the overall sampling number. I think this is linked with each other, especially for water vapour with its extreme

vertical and horizontal gradients. This objective would need further investigations for a final publication and is not sufficiently discussed by the only two sentences in section 3 (p5, line 25).

6) Further, the quantification of the number of years needed to compensate for the biases introduced by the intensive cloud-clearing for IR limb sounder, which is only briefly mentioned in the Summary and Conclusion section, would be a very interesting topic for more detailed analyses and would improve the quality of the manuscript.

**Minor comments:**

1) P2, line 23: Please, add a reference.
2) P3, line 3: Have you introduced CMAM30-SD already?
3) P4, line 7: Please, explain in more detail, why the horizontal gradients are that important and why a different retrieval scheme might fail.
4) P4, line 12: Are the ice cloud properties retrieved by a tomographic approach as well? Is a 2D or 3D tomographic approach, this should be mentioned.
5) This paper highlights technical aspects of the two sensors, so it might be helpful to include a minimum of instrumental details like field of view (FOV), vertical sampling (step size), retrieval resolution, and horizontal sampling. Otherwise the reader cannot judge on statements like 'the vertical resolution is good enough to resolve these model fields' without any reference.
6) Figure 1: please give some details why MIPAS fails to retrieve the most northern profiles. This looks like an artefact of the data processing. I would not expect strong horizontal gradients or clouds at these locations, like argued in the manuscript. Why fails MLS retrieval at some high northern latitudes as well? You should explain in more detail in the manuscript why and where retrieval my fail for both instruments.
7) Figure 3: Any explanation for the cold bias in the area of the south-pole for MIPAS? This is a bit confusing, because especially in the cold areas PSC are frequent and hamper proper retrievals for this part of the season, consequently I would expect a warm bias?!

**Technical comments:**

1) Figure 2, 3, and 4: A superimposed mean tropopause location will substantially improve the information content of the figures around the tropopause.
2) Figure 2: please, give a few more details in the figure caption. E.g. Za-Zr or Zr-Za?
3) Figure 5: The numbers on the x-axis are not well readable and should be sparser and separated.
4) Figure 6+7: Temperature biases should be presented with a scaling of T (e.g. K x 10). Then the significant temperature bias in the tropics will become more obvious. In addition, a tropopause location might help here as well.

**References:**

Millán, L. F., Livesey, N. J., Santee, M. L., Neu, J. L., Manney, G. L., and Fuller, R. A.: Case studies of the impact of orbital sampling on stratospheric trend detection and derivation of tropical vertical velocities: solar occultation vs. limb emission sounding, Atmos. Chem. Phys., 16, 11521-11534, https://doi.org/10.5194/acp-16-11521-2016, 2016.

Spang, R., Arndt, K., Dudhia, A., Höpfner, M., Hoffmann, L., Hurley, J., Grainger, R. G., Griessbach, S., Poulsen, C., Remedios, J. J., Riese, M., Sembhi, H., Siddans, R., Waterfall, A., and Zehner, C.: Fast cloud parameter retrievals of MIPAS/Envisat, Atmos. Chem. Phys., 12, 7135-7164, https://doi.org/10.5194/acp-12-7135-2012, 2012.

---

## Author Response (AR1)

**Response to reviewers**

We thank the reviewers for their comments. Below are our responses in blue.

In the course of responding to the reviewers, two main changes occurred: (1) a trend discussion was added and (2) we emphasize the lack of MLS penetration below the upper troposphere throughout the paper.

***Trend discussion***

Before the conclusions the following paragraphs will be added:

[revised manuscript text omitted]

Furthermore, we noticed that the Figures were not displaying the correct MLS pressure cut off. The revised versions showcase much better the lack of MLS penetration (as an example, the updated Figure 2 is shown below). Also, we superimpose a mean thermal tropopause derived from MERRA2 to Figure 2, 3, and 4.

[Figure]

Updated Figure 2

**Reviewer 1 comments**

This paper discusses the impact of non-uniform sampling of the MIPAS and MLS instruments on resulting zonally averaged data with particular emphasis on how data quality screening exacerbates biases. The authors describe how MIPAS and MLS have similar sampling biases but how MIPAS data is detrimentally affected by screening through the use of running MIPAS and MLS sampling through the CMAM30-SD CCM. While the methodology applied has some use and accurate in its basic conclusions, some of the more meaningful conclusions (e.g., impacts on ability to derive trends from different instruments in the UTLS) are not supported by analysis and thus speculative. Furthermore, the scope of this work is severely limited and the tone of the paper is seemingly self-serving. As such, I would recommend this work for publication only after additional work and major revision.

**Major Comments:**

Pg. 02, Line 29: "We emphasize that the results of this study refer only to the representativeness of the respective data, not to their intrinsic quality." This statement greatly detracts from the value of this work. This work makes very clear remarks regarding the impact of sampling biases on the ability to use certain kinds of data sets for trend analysis in the UTLS and how sampling biases will require longer data sets because of the added noise. However, the quality of the data sets that are used is a critical component to those kinds of analyses. Recent work in the 2014 Ozone Assessment or the SI2N effort (https://www.atmos-chem-phys.net/ special_issue284.html) have shown that sampling biases, while present in trend analyses, are not necessarily the greatest driver of trend uncertainty as data quality issues and instrument drifts are also present. Incorporating data quality into the calculation is necessary, as it is still possible for higher precision data with sampling biases to be more robust for trending work than lower precision data, though it ultimately depends on what those precisions and sampling impacts are.

This will be expanded to: We emphasize that the results of this study refer only to the representativeness of the respective data, not to their intrinsic quality. Their quality has been extensively evaluated in numerous data characterization and validation papers [i.e., Pumphrey et al., 2007; Read et al., 2007; Santee et al., 2007; Schwartz et al., 2008; Stiller et al, 2012; Hegglin et al., 2013; Raspollini et al, 2013; Neu et al 2014; Livesey et al., 2017; Sheese et al., 2017]. Furthermore, their long-term stability has also been studied [i.e., Nair et al., 2012; Eckert et al., 2014; Hubert et al., 2016; Hurst et al., 2016].

In addition, a new paragraph about trends will be added (see above) which will include the following sentence: Note that, when using real data, the effect of instrument noise upon trends will be negligible due to the vast number of MIPAS or MLS measurements associated with each monthly latitude bin. Drifts and long-term stability issues on these datasets [i.e., Eckert et al., 2014; Hubert et al., 2016; Hurst et al., 2016] will have to be corrected.

References:
Pumphrey et al. 2007: 10.1029/2007JD008723          Neu et al 2014:          10.1002/2013JD020822
Read et al 2007:          10.1029/2007JD008752          Stiller et al, 2012:          10.5194/amt-5-289-2012
Santee et al 2007:          10.1029/2007JD008721          Hegglin et al 2013:          10.1002/jgrd.50752
Schwartz et al 2008:          10.1029/2007JD008783          Raspollini et al 2013:          10.4401/ag-6338
Nair et al 2012:          10.5194/amt-5-1301-2012          Hubert et al 2016:          10.5194/amt-9-2497-2016
Eckert et al 2014:          10.5194/acp-14-2571-2014          Hurst et al 2016:          10.5194/amt-9/4447-2016

Sheese et al 2017:    10.1016/j.jqsrt.2016.06.026

Pg. 06, Line 32 and Pg. 07, Line 25: "These poor metrics imply that any trends derived at these pressure levels will also be impacted by quality screening induced biases: the magnitude of the trends will be affected because of the change in the slope, and the number of years of observations required to conclusively detect trends will considerably increase due to the noise associated with the worsening of the coefficients of determination (e.g., Millán et al., 2016)."

Biases and trends can be completely independent and the slopes that are referred to here are not trend slopes but correlation slopes. For example, a seasonally dependent dry bias in $H_2O$ as shown in Fig. 5 does not guarantee an induced bias in the trend values and so the impact of these sampling biases on trends is not directly addressed by this work. Furthermore, while it is true that increased noise in the data can result in requiring longer duration data records to determine significant trends, neither this study nor Millan et al. (2016) considers the attribution between data quality and non-uniform sampling. Without additional work to address the impacts of each of these factors, this entire statement is only speculative.

See discussion on trends above.

In general, the scope of this work is severely limited. The statement that an instrument's inability to see through clouds will result in less data is obvious while comparing two data sets to ascertain the impact on trends without considering data quality is neglectful. As it stands, this study has limited to no scientific value. However, the underlying concept of this work could be expanded albeit with simulated sampling and uncertainties. For example, the authors could simulate different sampling patterns from different orbit types and sampling frequencies for an expected limb sounder. Running these through model data would allow for comparisons of the impact of different potential sampling patterns. To further test potential data quality screening, utilizing model cloud fields or retrieval limitations of different observation techniques would create another variable to test rather than using the data quality screening of a specific instrument. Lastly, the authors could simulate differing data precisions or vertical resolutions from all of these sensitivity tests and incorporate the resulting uncertainties into trending analyses to determine the impact of all of these variables on the derived trends. This would provide a large solution space to test the impacts of potential measurement systems on their ability to be used for trending analyses in the UTLS region. While these suggestions are extensive and perhaps more comprehensive, some aspect(s) would be necessary to incorporate actual data quality into the authors' data quality screening to back up the claims regarding the impact on trend detection.

The main purpose of this study is to quantify the impact of screening biases using MIPAS and MLS as proxies for IR and MW limb sounder measurements, not to run a full Observing System Simulation Experiment (OSSE – the exercise the reviewer is essentially suggesting) to study different sampling patterns. That would make the paper much less focused. Furthermore, the MIPAS and MLS sampling patterns are so dense that the sampling bias is negligible for both (as shown here); that is, *the problem is not in the sampling pattern per se but rather in the inability to measure in cloudy scenes*, which leads to the "obvious" reduction in data. Hence no matter what sampling patterns are explored, the problem will still be there for IR based measurements. Furthermore, when dealing with real data, it will not only be a matter of the reduced number of measurements but also, the reduced representativeness of the average because clouds are correlated with the atmospheric state.

Moreover, an OSSE will require a threshold to determine which clouds affect the IR and the MW data. Such thresholds may be problematic to define and not actually useful in real-world scenarios. By using the screening applied to real data we circumvent this problem by using the criteria determined by the instrument teams to obtain the best data possible from their respective instruments.

Perhaps what is most troubling is the tone of the paper. It seems that the purpose of this paper is to act as a published reference for why a future MLS instrument must be used to ascertain long-term trends of trace gas species in the UTLS. The authors appear to go to great lengths to phrase the message as to assert the "superiority" of MLS measurements over MIPAS at every turn (i.e., pointing out all potential deficiencies in MIPAS data without mentioning any for MLS), and even go as far as attempting to undermine the usefulness of other current or future measurement systems as evidenced by the last two sentences in the paper. While this type of rhetoric is expected in a proposal, it is not suited for a scientific publication

We recognize the reviewer's concern, and hope that our changes detailed above (the inclusion of a discussion on the lack of MLS penetration), combined with our decision to delete the final sentence of the conclusion, alleviate them.

In addition, we will add in P4L5 (in the MIPAS IMK introduction): MIPAS IMK/IAA algorithm retrieves temperature and more than 30 species including $O_3$, $H_2O$, CO, CFCs, PAN, among many others.

**Minor Comments:**

Pg. 01, Line 23: "Further, satellite missions such as . . . have records that span more than a decade." This statement is phrased in such a way as to suggest that no ground station data have records that long. I would suggest revising. The sentence will be changed to: Further, like many ground-based data sets, satellite mission such as …

Pg. 02, Line 09: "They concluded that coarse non-uniform sampling leads to nonnegligible biases . . ." Biases in what? Is that data biased from the sampling or are the analysis methods not conducive to using data with non-uniform sampling? The sentence will be changed to: They concluded that coarse non-uniform sampling leads to non-negligible zonal mean biases.

Pg. 02, Line 13: "They found that coarse non-uniform sampling patterns can induce significant errors in the magnitudes of inferred trends . . ." Again, is this a flaw in the data or the analysis method? This will be changed to: They found that coarse non-uniform sampling patterns can induce significant errors in the magnitudes of trends inferred directly from monthly zonal means, …

Pg. 04, Line 12: "MLS measures around 3500 vertical scans daily, providing nearglobal (82S to 82N) observations." Please also include the geospatial sampling characteristics of the MIPAS instrument in its description. It is in P3 L22 of the original manuscript "MIPAS measured around 1350 vertical scans daily, providing global observations." Thus, no changes have been made in the revised manuscript to address this comment.

Pg. 04, Line 20: "Further, we used the vertical grid of the CMAM30-SD fields; that is, we assume that MIPAS and MLS vertical resolution is good enough to resolve these model fields, at least in the upper troposphere / lower stratosphere (UTLS)." What are the vertical resolutions of the model and instruments in the UTLS?

The vertical resolution of the model is in the original manuscript (P3 L15). For MIPAS and MLS, the vertical resolution will vary by specie but overall, they are between 3 and 4 km. Taking this into account, the sentence will be changed to: Further, we used the vertical grid of the CMAM30-SD fields; that is, the impact of the vertical resolution of the measurements is not taken into account. However, note that, in this case, both instruments have similar vertical resolutions in the upper troposphere / lower stratosphere (UTLS), varying overall from 3 to 4 km.

Pg. 06, Line 31 and Pg. 07, Line 24: "The biases for O3 and HNO3 oscillate between -10.   Unfortunately, we couldn't address this comment as it seems to us that it is incomplete.

Figure 5: It would be better to change the X-axis label interval on the time-series plot to every 5 years.
We changed it to every two years and increased the font size.

Figures 6 and 7: There is a lot of unnecessary white space in some of these plots, though given the desire to maintain consistent axes ranges between the two I can see why Figure 6 has so much white space. That having been said, I still think some reductions can be made to make the results easier to see.
The x-ranges have been reduced as much as they can be without cutting out any of the lines shown.

Additionally, for whatever reason the line thicknesses appear the same between Figs. 6 and 7 when zoomed out but are much thinner in Fig. 6 when zoomed in. They will be the same thickness in the new version.

Lastly, what is the bottommost pressure level on the Y-axis?
The bottommost level and the top pressure level are 400 and 50 hPa, these levels will be added to the figures.

**Reviewer 2 comments**

The paper is dedicated to the characterization of sampling biases in infrared and microwave limb sounding instruments, with MIPAS and MLS taken as examples. The paper is a continuation of a series of publications on characterization of sampling biases. The new aspect is analyzing the influence of quality screening on data representativeness.

**MAJOR COMMENTS**

1) It is stated in the abstract that " analysis of long-term time series reveals that these additional quality screening biases may affect the ability to accurately detect upper tropospheric long-term changes using such data" (similar statements are on page 6 and in conclusions) However, the performed analyses are insufficient for such statement. It is rather expected that the screening of cloudy conditions results in biased estimates, and that the variability might not be represented properly. However, biased estimates, not perfect correlation coefficient with the full time series and R2 do not necessary imply that the long-term trends are inaccurate. Furthermore, if the sampling patterns do not change over time, a large part of sampling uncertainty can be removed in the trend analysis by consideration of deseasonalized anomalies. In order to make such statement on ability of accurate trend detection, the authors should perform trend analysis using the full and sub-sampled datasets and support their statements by quantitative estimates. Another, a simpler solution, is to remove these abovementioned statements on ability to accurately detect trends from the manuscript.

See discussion on trends above.

If the authors will decide to extend the analyses, it would be also interesting to investigate the influence of sampling patterns on ability to reproduce the natural cycles.

We decided not to expand the manuscript upon the ability to reproduce natural cycles because we believe that is outside the scope of the current paper.

2) The value of the paper will be increased significantly, if the presented analyses of sampling biases using the modelled data are enhanced with comparison of real experimental data from MIPAS and MLS. Such analyses would illustrate whether the observed biases are explained by sampling patterns.

After careful consideration, we decided not to include a comparison of the real data, because such comparisons will suffer from the fact that we do not know the truth and because such comparisons can be found in several validation papers. The fact that extensive validation of these data sets has been documented in previous validation papers is now noted in the revised manuscript.

**MINOR COMMENTS**

1) P.4, L.5 : Please write the version of the IMK/IAA processor. We will add: in particular version 5.

2) P.2 L.11: The sampling uncertainty has been also discussed in (Sofieva et al., 2014). In this paper, the authors analysed the sampling biases for 6 satellite instruments and proposed a parameterization of sampling uncertainty in monthly zonal mean data.

Sofieva, V. F., Kalakoski, N., Päivärinta, S.-M., Tamminen, J., Laine, M. and Froidevaux, L.: On sampling uncertainty of satellite ozone profile measurements, Atmos. Meas. Tech., 7(6), 1891–1900, doi:10.5194/amt-7-1891-2014, 2014. 3)

We will modify that section as follows: For the limb sounding technique, Sofieva et al., (2014) estimated the sampling biases in zonal mean ozone profiles from six limb-viewing satellite instruments and proposed a simple parameterization to estimate them. Toohey et al., (2013) characterized the sampling bias for $H_2O$ and $O_3$ …

Figure 5: Please use more distinct colors in scatter plots.
The colors were changed, see below:

[Figure]

The caption will be changed accordingly: The dashed gray lines are the 1:1 line, and the solid lines are the linear best fits, whose slopes are given.

**Reviewer 3 comments**

The paper by Millan et al. describes effects of sampling and quality screening for two limb sounding instruments (MLS and MIPAS). Based on CCM data the analysis shows significant biases in the zonally averaging of various trace gases and temperature distribution, especially for a MIPAS-like instruments with its cloud-induced quality screening. Biases in the zonal means as well as for trend analyses in the upper troposphere can be quite large, which is an important fact and needs consideration in any trend analysis.

**General comments:**

There are two major concerns regarding the manuscript from Millan et al.:

a) The study highlights a very specific caveat of sampling and screening issues of space borne instruments. The whole approach is very technical, investigating a very obvious problem of the IR limb measurements due to cloud screening, which can have a severe effect on the results especially if you like to analyse trends in the very cloudy upper tropical troposphere. Although this effect is of interest for ACP community, these kind of analysis is not really suited for ACP. The whole manuscript looks more like a technical study or a technical note (for example as prepared for NASA or ESA studies for future satellite sensors) and is not sufficient for a full publication in a journal like ACP. A more in-depth study of the MW/IR limb caveats on trend analyses cloud be publishable in a more technical and instrument orientated journal like Atmospheric Measurement Techniques.

As the referee notes, determination of trends is a critical issue of great current concern. While the effects of clouds on IR measurements are indeed well known, the impact of removing cloud-affected profiles on biases in zonal means and the trends estimated from them has not been quantified before. Hence, we believe that this paper is appropriate for ACP because its results directly impact the broad UTLS community.

b) To my mind, this paper does not contribute a substantial add-on to a former study of the authors. The paper by Millan et al. (2016) in ACP describes most of the general effects of sampling pattern, which can be quite similar effects to the quality screening, in the context of a much more detailed study on atmospheric trend analyses. My strong impression is that the actual manuscript misses new concepts and methods and just repeats parts of the Millan et al. (2016) analyses with another combination of satellite instruments.

The new addition is the quantification of the screening bias *on top of* the sampling bias. Previous studies (Toohey et al., 2013; Sofieva et al., 2014; Millán et al., 2016) focused on the sampling bias while disregarding the effects of data quality screening. As shown in Figure 3 and 4 of the paper, the effects of the screening biases are not insignificant and need to be known and quantified. A different combination of satellite instruments was essential to this study because the idea was to take data sets with comparable sampling biases and show how they diverge when screening biases are taken into account.

P2 L17 will be expanded:  However, none of these studies have quantified the additional biases introduced through quality screening of the measurements, that is, this study isolates and quantifies another source of uncertainty in averaged data.

**Major comments:**

1) P4, line 29: Although the screening is a fundamental issue of the study, there are no details presented for MLS (but some for MIPAS). Please, at least summarize the main topics of Livesey et al. (2006). In addition, more details on the cloud screening of MIPAS would be helpful as well (see next comment).

We will add: The screening procedure applied to MLS data follows the guidelines detailed by Livesey et al. (2017): We only use data within the specified pressure ranges; we neglect profile points for which the precision is negative (which indicates that the retrievals are influenced by the a priori); we avoid profiles for which the "Status" field is an odd number (which indicates operational abnormalities or problems with the retrievals); and we only use profiles for which the "Quality" and "Convergence" fields are within the specified thresholds. The "Quality" field describes the degree to which the measured radiances have been fitted by the retrieval algorithm and the "Convergence" field is a ratio of the fit achieved at the end of the retrieval process to the value predicted at the previous step.

2) P5, line 1: Please note somewhere here that the MIPAS cloud-screening applied in the IMK product is a very conservative approach with respect to cloud effects in the measured spectra. This might have effects on the presented analysis. Especially, in the tropical upper troposphere high water vapour abundance will result in false-positive cloud detections (e.g. Spang et al. 2012) in the IMK product and consequently an artificial overestimation in the biases, where other MIPAS level 2 processors might do a better job. This objective should be addressed in the manuscript.

The following sentence will be added in P5 line 28: We note that the cloud screening procedure of the IMK/IAA algorithm is conservative with respect to those of other MIPAS processors [Spang et al., 2012]. On the face of it, it appears that this causes an unnecessarily large sampling bias which could be avoided by using a less restrictive cloud screening threshold value. The purpose of a conservative cloud screening procedure is to guarantee that the measurements passing the cloud screening are indeed unaffected by any cloud signal in the spectra. Cloud signals can lead to systematic retrieval errors, which are correlated to the state of the atmosphere, hence, the sampling biases would merely be replaced by retrieval biases. This, we think, is worse, because then both the parent data and the zonal averages would be affected, while with a conservative screening only the averages are affected but not the parent data which survive the screening.

3) The disadvantages of the cloud screening in the tropical UT for the IR measurements are highlighted frequently in the manuscript and are presented by different types of analyses, but with only very limited further information content. In contrast, the caveat of MW sounders to retrieve trace gases below ~300 hPa at any latitude is only briefly mentioned. Studies for future ESA missions, like the PREMIER report for mission selection (http://projects.knmi.nl/capacity/PREMIER/SP1324-3_PREMIERr.pdf), show nicely the excellent synergy by a combination of IR and MW (s. Fig. 4.2 of the report), where IR-limb still allows measurements in the mid-troposphere, where MW-limb fails to measure.

See MLS lack of penetration section above.

4) To my mind the trend analysis section needs to be improved. Although MIPAS tackles to deliver good trend analyses in tropical UT, this is a shortened result. Have you tested if the biases are better, if the

trend analyses are restricted to longitude regions where convection is less pronounced (e.g. mid-Pacific, Atlantic regions, excluding the warm pool region, central Africa and South America, where deep convection and cloud-coverage is most pronounced). Local trends in the UT/MT in tropical and subtropical areas would be valuable in principle.

See trends section above.

The reviewer is correct in pointing out that trends restricted to longitude regions where convection is less pronounced will be less affected (or barely affected) by cloud screening biases. For example, the following figure shows the yield at 200hPa for MIPAS and MLS water vapor and $O_3$ as well as the trends computed in the mid-Pacific where the MIPAS yields are less affected. As can be seen, the impact of the quality screening upon the trends in these region is minimal. In the paper we will add: Furthermore, MIPAS trend analysis can be restricted to regions less affected by deep convection (for example, the mid tropical Pacific) to minimize the quality screening effects.

[Figure]

Figure: (right) Yield maps, (left) trends computed in the mid-pacific area shown in the maps for $O_3$ and $H_2O$.

5) I am also wondering why the authors argue that most of the biases are not caused by a lack of observations in convective (cloudy) areas in the tropics but are caused by a reduction in the overall sampling number. I think this is linked with each other, especially for water vapour with its extreme vertical and horizontal gradients. This objective would need further investigations for a final publication and is not sufficiently discussed by the only two sentences in section 3 (p5, line 25).

The sentence will be expanded as follows:
Although this resembles the expected dry bias in clear-sky tropospheric infrared measurements (e.g., Sohn et al., 2006; Yue et al., 2013) — that is, the fact that infrared instruments cannot measure cloudy regions where $H_2O$ is high, resulting in a dry bias — **the biases shown here are due to a combination of two factors: (1) high H2O values associated with deep convection (the screened-out locations might not necessarily be cloudy in the model fields however they are for the most part in the tropics, in regions of high H2O values, see Figure 1 for an example) and (2) due to the reduced sampling frequencies.** Note that this also applicable to other parameters …

6) Further, the quantification of the number of years needed to compensate for the biases introduced by the intensive cloud-clearing for IR limb sounder, which is only briefly mentioned in the Summary and Conclusion section, would be a very interesting topic for more detailed analyses and would improve the quality of the manuscript.

See trends section above.

**Minor comments:**
1) P2, line 23: Please, add a reference. We will add: [e.g., Livesey and Read (2000), Carlotti et al. (2001, 2006), Kiefer et al. (2010), Castelli et al. (2016)]. ]

[revised manuscript text omitted]

---

## Referee Report (RR1)

The authors have made changes to the manuscript to remove potential biases and improve the overall message as well as include a section about the impact on trends. However, it appears that only a limited amount of work went into this new analysis and the methodology applied is far from robust. Additionally, while the addition of a section on trends was necessary in order to try to substantiate the claims made in the paper, the lack of any additional work means that the scope of this manuscript remains very limited as stated in the previous review. As such, I would agree with Reviewer 3 in that this work, subject to additional corrections outlined below, would be better suited for publication in AMT rather than ACP. However, I will defer the decision on which journal this manuscript belongs in to the editor.

**Comments:**
Pg. 07, Ln. 22: "These poor metrics imply that any trends derived at these pressure levels will also be impacted by quality screening induced biases: the magnitude of the trends will be affected because of the change in the slope, and the number of years of observations required to conclusively detect trends will considerably increase due to the noise associated with the worsening of the coefficients of determination (e.g., Millán et al., 2016)."

As stated in the previous review, changes in the correlation slope do not necessarily mean that the trends are affected. In fact, a comparative analysis of figures 7 and 8 show this. The slopes for ozone in the tropics in Fig. 7 show large departures from 1 (just as much as water vapor) between 100 and 300 hPa and yet no significant differences in the trends in this region. Similarly, the relationship between the coefficients of determination and the years of observations required to detect trends is also questionable. The coefficients for ozone appear to progressively degrade below 100 hPa along with an increase in the number of years for trend detection. However, the number of years also increases significantly above 100 hPa with no corresponding significant change in any of the correlation parameters. Granted, the ozone trends are small here but they are also small for water vapor and there does not appear to be any degradation in trend results there.

The new trend section states that the impact of sampling bias on the water vapor / ozone trends are up to 80% / 20% and that the number of years for robust trend detection changes by 150 / 40 years respectively. These changes appear to be the largest possible, which are well into the middle troposphere. However, the manuscript only makes mention of results in the upper troposphere. In the case of ozone, restricting the quoted analysis results to the upper troposphere yields negligible differences in trend results and much smaller changes in years required.

In general I am pleased that the authors decided to include a section in this work on trend analysis at the request of all of the reviewers. However, I am disappointed in the limited amount of work that went into this. For starters, the resulting trends do not have any uncertainties, making it impossible to determine if the changes in trends between sampling patterns are statistically significant. Second, the choice of regression model for trend determination (assuming it to be the one used in Millán et al., 2016) is far too simple to yield robust results, particularly for ozone or water vapor in the troposphere. The authors should include additional dynamical proxies that tend to have large influences on atmospheric species via transport to better characterize variability (common choices are related to the QBO, ENSO, volcanism, the Asian monsoon, and eddy heat flux). This simple choice of trend model will have a detrimental

impact on the computation of the number of years of observations necessary for robust trend detection. The calculation from Tiao et al. and Weatherhead et al. is strongly dependent upon the nature of the residuals and such a simple trend model in the presence of strong dynamical influences will artificially inflate the resulting values. Unfortunately, including additional proxies in the regression model will require a rederivation of this equation as the one derived in those references and used by the authors is only valid for the simplified regression model.

---

## Author Response (AR2)

**Response to reviewer**

We thank the reviewer for his/her comments. Below are our responses in blue.

Reviewer comments

The authors have made changes to the manuscript to remove potential biases and improve the overall message as well as include a section about the impact on trends. However, it appears that only a limited amount of work went into this new analysis and the methodology applied is far from robust. Additionally, while the addition of a section on trends was necessary in order to try to substantiate the claims made in the paper, the lack of any additional work means that the scope of this manuscript remains very limited as stated in the previous review. As such, I would agree with Reviewer 3 in that this work, subject to additional corrections outlined below, would be better suited for publication in AMT rather than ACP. However, I will defer the decision on which journal this manuscript belongs in to the editor.

Comments:

Pg. 07, Ln. 22: "These poor metrics imply that any trends derived at these pressure levels will also be impacted by quality screening induced biases: the magnitude of the trends will be affected because of the change in the slope, and the number of years of observations required to conclusively detect trends will considerably increase due to the noise associated with the worsening of the coefficients of determination (e.g., Millán et al., 2016)."
As stated in the previous review, changes in the correlation slope do not necessarily mean that the trends are affected. In fact, a comparative analysis of figures 7 and 8 show this. The slopes for ozone in the tropics in Fig. 7 show large departures from 1 (just as much as water vapor) between 100 and 300 hPa and yet no significant differences in the trends in this region. Similarly, the relationship between the coefficients of determination and the years of observations required to detect trends is also questionable. The coefficients for ozone appear to progressively degrade below 100 hPa along with an increase in the number of years for trend detection. However, the number of years also increases significantly above 100 hPa with no corresponding significant change in any of the correlation parameters. Granted, the ozone trends are small here but they are also small for water vapor and there does not appear to be any degradation in trend results there.

The reviewer is correct to point out that we overstated in P7 line 22; that sentence will be changed to: These poor metrics suggest that any trends derived at these pressure levels might also be impacted by quality screening induced biases.

By the way, the increase in the number of years above 100 hPa for $O_3$ is due to the minor changes in the estimated trends at those pressure levels (the color lines can barely be seen on top of the black line), which leads to the significant changes in the number of years. We feel that no change in the text is needed with regard to this point, because even when using the raw model fields more than a hundred years are required.

The new trend section states that the impact of sampling bias on the water vapor / ozone trends are up to 80% / 20% and that the number of years for robust trend detection changes by 150 / 40 years respectively. These changes appear to be the largest possible, which are well into the middle troposphere. However, the manuscript only makes mention of results in the upper troposphere. In the case of ozone, restricting the quoted analysis results to the upper troposphere yields negligible differences in trend results and much smaller changes in years required.

The reviewer is correct to point out that we were using the largest possible changes; we modified the text to reflect the changes at 200 hPa. P7 line 32 was changed to: MIPAS trends are impacted because of the large percentage of measurements screened out below 100 hPa, which introduces non-negligible artifacts (for example, 0.8% decade $^{-1}$ for $H_2O$ at 200 hPa versus 1.8% decade $^{-1}$ when all available measurements are used).

Furthermore, in P8L7 we will add: As shown, with the MIPAS screened fields additional years are required for robust trend detection (up to a total of ~120 years for $H_2O$ and up to ~25 years for $O_3$ at 200 hPa versus 50 years and 18 years, respectively)

In general I am pleased that the authors decided to include a section in this work on trend analysis at the request of all of the reviewers. However, I am disappointed in the limited amount of work that went into this. For starters, the resulting trends do not have any uncertainties, making it impossible to determine if the changes in trends between sampling patterns are statistically significant. Second, the choice of regression model for trend determination (assuming it to be the one used in Millán et al., 2016) is far too simple to yield robust results, particularly for ozone or water vapor in the troposphere. The authors should include additional dynamical proxies that tend to have large influences on atmospheric species via transport to better characterize variability (common choices are related to the QBO, ENSO, volcanism, the Asian monsoon, and eddy heat flux). This simple choice of trend model will have a detrimental impact on the computation of the number of years of observations necessary for robust trend detection. The calculation from Tiao et al. and Weatherhead et al. is strongly dependent upon the nature of the residuals and such a simple trend model in the presence of strong dynamical influences will artificially inflate the resulting values. Unfortunately, including additional proxies in the regression model will require a rederivation of this equation as the one derived in those references and used by the authors is only valid for the simplified regression model.

We agree with the reviewer that a more robust trend analysis would be better, however it is outside the scope of this study. The trend section will be modified to include: Furthermore, a more robust trend analysis that includes the influence of dynamical variables, such as the quasi-biennial oscillation and El Niño–Southern Oscillation, will help reduce the number of years required to statistically detect such trends. In addition, 
[revised manuscript text omitted]
 (for example, 0.8% decade$^{-1}$ for $H_2O$ at 200 hPa versus 1.8% decade$^{-1}$ when all available measurements are used); MLS trends are impacted because of the reduced vertical resolution, which limits its usefulness to the upper troposphere and above. Note that the impact of quality screening on MIPAS trends can be mitigated by using a regression model similar to the ones used by Bodeker et al. (2013) and Damadeo et al. (2014). These models have been shown to mitigate the effects of the non-uniform temporal, spatial and diurnal sampling of solar occultation satellite measurements. Furthermore, a more robust trend analysis that includes the influence of dynamical variables, such as the quasi-biennial oscillation and El Niño-Southern Oscillation, will

help reduce the number of years required to statitiscally detect such trends. In addition, MIPAS trend analysis can be restricted to regions less affected by deep convection (for example, the mid tropical Pacific) to minimize the quality screening effects.

The estimated number of years required to definitively detect these trends is also shown in Figure 8. These estimates were computed assuming a trend model similar to the one described by Tiao et al. (1990), Weatherhead et al. (1998), and Millán et al. (2016), with a seasonal mean component represented by the monthly climatological means. As shown, with the MIPAS screened fields additional years are required for robust trend detection (up to a total of $\sim$120 years for $H_2O$ and up to $\sim$25 years for $O_3$ at 200 hPa 
[revised manuscript text omitted]